# Synergies Radiotherapy-Immunotherapy in Head and Neck Cancers. A New Concept for Radiotherapy Target Volumes—“Immunological Dose Painting”

**DOI:** 10.3390/medicina57010006

**Published:** 2020-12-23

**Authors:** Camil Ciprian Mireştean, Anda Crişan, Călin Buzea, Roxana Irina Iancu, DragoşPetru Teodor Iancu

**Affiliations:** 1Department of Oncology and Radiotherapy, University of Medicine and Pharmacy of Craiova, 200349 Craiova, Romania; mc3313@yahoo.com (C.C.M.); anda_crisan2005@yahoo.com (A.C.); 2Euroclinic Center of Oncology Iaşi, 700110 Iaşi, Romania; 3Department of Radiotherapy, County Clinical Emergency Hospital Craiova, 200642 Craiova, Romania; 4National Institute of Research and Development for Technical Physics, 700050 Iaşi, Romania; calinb2003@yahoo.com; 5Department of Radiology, “Prof. Dr. Nicolae Oblu”, Clinical Emergency Hospital, 700309 Iaşi, Romania; 6Department of Oral Pathology, “Gr. T. Popa” University of Medicine and Pharmacy, 700115 Iaşi, Romania; dt_iancu@yahoo.com; 7Department of Oncology and Radiotherapy, “Gr. T. Popa” University of Medicine and Pharmacy, 700115 Iaşi, Romania; 8Department of Clinical Laboratory, “St. Spiridon” Emergency Hospital, 700111 Iaşi, Romania; 9Department of Radiotherapy, Regional Institute of Oncology, 700483 Iaşi, Romania

**Keywords:** radiotherapy, immunotherapy, ab-scopal, re-irradiation, head and neck cancers

## Abstract

The combination of immune checkpoint inhibitors and definitive radiotherapy is investigated for the multimodal treatment of cisplatin non-eligible locally advanced head and neck cancers (HNC). In the case of recurrent and metastatic HNC, immunotherapy has shown benefit over the EXTREME protocol, being already considered the standard treatment. One of the biggest challenges of multimodal treatment is to establish the optimal therapy sequence so that the synergistic effect is maximal. Thus, superior results were obtained for the administration of anti-CTLA4 immunotherapy followed by hypofractionated radiotherapy, but the anti-PD-L1 therapy demonstrates the maximum potential of radio-sensitization of the tumor in case of concurrent administration. The synergistic effect of radiotherapy–immunotherapy (RT–IT) has been demonstrated in clinical practice, with an overall response rate of about 18% for HNC. Given the demonstrated potential of radiotherapy to activate the immune system through already known mechanisms, it is necessary to identify biomarkers that direct the “nonresponders” of immunotherapy towards a synergistic RT–IT stimulation strategy. Stimulation of the immune system by irradiation can convert “nonresponder” to “responder”. With the development of modern techniques, re-irradiation is becoming an increasingly common option for patients who have previously been treated with higher doses of radiation. In this context, radiotherapy in combination with immunotherapy, both in the advanced local stage and in recurrent/metastatic of HNC radiotherapy, could evolve from the “first level” of knowledge (i.e., ballistic precision, dose conformity and homogeneity) to “level two” of “biological dose painting” (in which the concept of tumor heterogeneity and radio-resistance supports the need for doses escalation based on biological criteria), and finally to the “third level“ ofthe new concept of “immunological dose painting”. The peculiarity of this concept is that the radiotherapy target volumes and tumoricidal dose can be completely reevaluated, taking into account the immune-modulatory effect of irradiation. In this case, the tumor target volume can include even the tumor microenvironment or a partial volume of the primary tumor or metastasis, not all the gross and microscopic disease. Tumoricidal biologically equivalent dose (BED) may be completely different from the currently estimated values, radiotherapy treating the tumor in this case indirectly by boosting the immune response. Thus, the clinical target volume (CTV) can be replaced with a new immunological-clinical target volume (ICTV) for patients who benefit from the RT–IT association (Image 1).

## 1. Introduction

Radiation therapy in association with cisplatin-based chemotherapy or cetuximab, a monoclonal anti-epidermal growth factor receptor (EGFR) antibody, is the cornerstone of locally advanced head and neck cancers (HNC), the concurrent treatment demonstrating a synergistic potential. Currently, definitive concurrent radio-chemotherapy using the intensity modulated radiotherapy (IMRT) technique is the therapeutic standard for this stage of the disease. The combination of immune checkpoint inhibitors and definitive radiotherapy is investigated for the non-eligible cisplatin patients in the treatment of locally advanced HNC. For recurrent and metastatic HNC, immunotherapy has shown benefit over the EXTREME protocol, isalready considered as standard treatment. Recently, immunotherapy has been shown to be safe and potentially effective even in neo-adjuvant settings of head and neck cancers. One of the biggest challenges in multimodal treatment is to establish the optimal therapy sequence so that the synergistic effect is maximal. Analyzing the results of preclinical studies, we consider the maximum potential benefit of the combination of immunotherapy and radiotherapy in locally advanced, recurrent and metastatic cases, situations in which the primary tumor, lymph node or visceral metastasis remains “in place”. For this reason, the neo-adjuvant immunotherapy for HNC will not be the subject of this analysis [1,2,3,4].

## 2. Radiotherapy-Immunotherapy Synergy—From Mechanisms to Clinical Practice

The tumor-host relationship evolves during the development of the tumor from the moment that the first cell is recognized and destroyed by the immune system to a moment of balance and coexistence of tumor and immune elimination mechanisms until the “escape” occurs. The “Immune escape” is associated with the moment in which the tumor microenvironment becomes immunosuppressive. Thus, by reducing the antigen’s presentation, the tumor is “hidden” from the immune system, the balance being inclined towards the mechanisms of tumor proliferation. The role of radiotherapy is to “expose” the tumor of the adaptive and innate immune system [5].

A calcium-binding protein (calreticulin) is released from the endoplasmic reticulum under the influence of irradiation, being involved in the anti-phagocytosis mechanisms of CD4lymhocites. Stimulation of destroyed cells clearance using antigen-presenting cells (APC), stimulation of T cells and activation of high mobility group box 1 (HMGB1), but also activation of major histocompatibility complex (MHC-1) which facilitates the recognition of tumor cells by cytotoxic T lymphocyte are mechanisms by which the immune system is stimulated by irradiation.

In clinical practice, a radiation-stimulated increase in APC may open new horizons for the synergistic effects for the association of irradiation with CAR-T cell therapy. By inducing DNA damage, irradiation can generate antigens that facilitate immune recognition, the potential of this effect being increased especially in cells with DNA repair, and may potentiate an associated radiotherapy–immunotherapy (RT–IT) treatment. Because the effect of neoantigen generation is also associated with healthy tissue, the RT–IT treatment may also increase the rate of side effects, not just the tumoricidal effect. By activating the stimulatory pathway of interferon genes (STING), radiation therapy modulates the DNA-mediated immune response. The STING pathway is involved in radiation-induced interferon-1 immunity in malignant melanoma and colorectal cancer. The STING pathway also benefits from targeted medication, currently proposed in clinical trials. Activation of the STING and interferon-1 pathway may lead to the recruitment of myeloid-derived suppressor cells resulting in immunosuppression and radio-resistance [6,7].

The immune-modulatory effect of irradiation is also manifested in the case of tumor microenvironment by inhibition of the transferring growth factor β (TGF-β). There is preclinical evidence that irradiation associated with TGF-β inhibition has led to increased T-cell priming with the potential to improve patient prognosis by adding anti-PD-1 therapy. One of the biggest challenges in combining two or more therapeutic methods is to establish the optimal therapeutic sequence in order to obtain a maximal synergistic effect.Preclinical studies have shown a correlation of the optimal treatment sequence with the type of chosen immunotherapy. Thus, superior results were obtained for the administration of anti-CTLA4 therapy followed by hypofractionated radiotherapy, but the anti-PD-L1 therapy demonstrates the maximum potential of radio-sensitization of the tumor in case of concomitant administration. The synergistic effect of RT–IT was also proven in the clinical practice by the KEYNOTE-001 trial, in which case radiotherapy was administered before pembrolizumab therapy. In the case of durvalumab, with a 14 days cutoff after the completion of radio-chemotherapy, a superior progression-free survival (PFS) was demonstrated for patients with lung cancer in the case when immunotherapy was administered at a shorter interval after radio-chemotherapy. A still unresolved dilemma remains the optimal fractionation scheme, the controversy being whether standard fractionation induces a synergistic effect in association with immunotherapy at least as intense as hypo-fractionation. The preclinical evidence that evaluated the effect of fractionation isso far contradictory. In the case of malignant melanoma, a dose of 15Gy induced more tumor-infiltrating T cells than a fractionated dose, while in the case of ab-scopal effect induction, it was observed in a preclinical model only in the case of breast cancer fractional irradiation. The T lymphocytes are inevitably included in the irradiation field during radiation treatment, and prolonged irradiation with standard fractionation may induce lymphopenia. It is highlighted that even a single fraction of irradiation can induce the death of lymphocytes from the irradiated anatomical region. The poor prognosis of the patient diagnosed with non-small long cell carcinoma (NSCLC) and nasopharyngeal cancer associated with lymphopeniahasalready been demonstrated. Irradiation technique and elective nodal irradiation may affect the amplitude of lymphopenia. Evidence that proton beam therapy has generated an enhanced immunotherapy potentiating effect may suggest that favorable dosimetry plays an essential role in this phenomenon by majorly reducing healthy tissue irradiation and by reducing the risk and severity of radiation-induced lymphopenia. Nodal elective irradiation did not affect tumor control in the case of animal models but reduced the number of infiltrating CD8 + T cells. Actually, it has not been proven yet whether elective nodal irradiation will sabotage tumor immunity or will have a potentiating effect in the case of RT–IT association [7,8,9,10,11].

## 3. Biomarkers for Head and Neck Cancers Radio-Immunotherapy

Identifying an ideal biomarker for RT–IT synergy remains a research topic for the future. Even if currently tumor mutation burden, DNA repair deficiency, tumor-infiltrating lymphocytes and more recently, gut microbiome are associated with the response to immunotherapy, no universal biomarker has been identified for the therapeutic association. In the case of chemotherapeutic agents and of target therapies, the radio-sensitizing effect is indisputable, but for the association of IT-RT, there are still limited data available. Thus, inhibition of poly-ADP ribose polymerase (PARP) and an upregulated PD-L1 expression have an immunosuppressive effect. p53, known as a demonstrated radio-sensitivity modulator, also proved the possibility of PD-L1 modulation. Jang et al. demonstrate the link between radio-sensitivity and PD-L1 expression tested in patients with invasive breast cancer using CD274 mRNA expression as a surrogate for PD-L1 expression. The authors consider PD-L1 expression as an important factor involved in the prediction of outcome, proposing the use of PD-L1 as a decision biomarker in RT–IT administration.

Starting from the idea that overexpression of PD-1 and PD-L1 is associated with an unfavorable response to radiotherapy and given the increased radio-sensitivity of HNC associated with human papillomavirus (HPV) infection, Lyu et al. aim to assess whether the difference in radio-sensitivity is related to PD-1/PD-L1 expression in tumors with different HPV status. The authors identify PD-L1 expression is increased in the tumor relative to normal tissue, and PD-L1 was correlated with PD-1 expression, HPV/p16 cancers being characterized by increased PD-1 expression. PD-L1 expression and PD-expression 1 were both identified as predictors of radio-sensitivity. PD-1 expression was associated with better recurrence-free survival (RFS), and among high PD-L1 patients, cases with radio-resistance and negative HPV/p16 had a lower overall survival (OS), which isconsidered the group of patients which could have the greatest benefit from RT–IT synergy [1,4,9].

Quan and colleagues assessed the PD-L1 expression and TILs level in 96 p16-negative and p16-positive HNC patients using immunohistochemistry, and the correlations between PD-L1 expression, TIL and p16 status were analyzed. The authors did not identify any differences in PD-L1 expression for p16 positive and negative tumors. P16 and PD-L1 expression were not correlated with OS and TIL, and CD8 were both identified as independent and favorable prognostic factors [10].

The expression of PD-L1 and CD8 TILs have been shown to be independent prognostic factors regardless of resection status, lymph node invasion, extra-capsular invasion in patients with HNC treated by surgery followed by chemo-irradiation. In the case of chemo-radiotherapy, the presence of CD3, CD8 TILs and PD-L1 expression were identified as prognostic factors [11,12].

Tumor mutational burden (TMB) and T-cell inflamed gene expression profile (GEP) also demonstrated the potential of biomarker, following a multivariate analysis of data from a retrospective study, the response to the treatment being superior if both values of the two biomarkers are increased. In HNC HPV+, the benefit of using PD-1/PD-L1 inhibitors was higher for HPV+ patients than in HPV− patients and HPV+ status has been associated with an increased value of T cell infiltration, cytolytic-immunity profile and an inflamed immune microenvironment, which justifies the possible benefit of immunotherapy. The authors propose HPV status as a biomarker independent of PD-L1 and TMB expression for response to PD-1/PD-L1 inhibitor therapy [13].

The relationship between the increased neutrophil/lymphocyte ratio (NLR) and the unfavorable prognosis has been demonstrated in most cancers, including HNC. The association of NLR with platelet-to-lymphocyte ratio (PLR) increases the accuracy of the prognosis. In the case of HNC, a study that included only radiation-treated patients demonstrated the prognostic value of NLR for hypo-pharyngeal or nasopharyngeal laryngeal cancers, but not for oropharyngeal cancers. Szilasi et al. consider a threshold value of 3.9 for NLR can be considered an independent risk factor for 5-year survival in HNC. Evaluating NLR at the beginning and in the first month during treatment in patients who received immune checkpoint inhibitors for advanced cancer, Li and colleagues evaluated the prognostic value of NLR and the dynamics of this value during treatment. The authors concluded that the change in NLR over time is a nonlinear predictor of the prognosis and that the longest survival is obtained for patients who showed a slow decrease in NLR, while an accelerated dynamics of this value was associated with unfavorable prognosis. NLR has already demonstrated the potential to predict the response to immunotherapy in lung cancer; NLR ≤ 5, or <2.5 according to other authors, being associated with a favorable response to immunotherapy. Ren et al. also correlated elevated CD3+ and CD8+/CD28+ T cell values with NLR <2.5, but by analyzing NLR in association with TMB, the predictive value of NLR becomes insignificant for TMB ≤10.

Even in the case of immunotherapy that inhibits PD-1/PD-L1, the biomarkers identified until now are different. For example, CD8 and Treg T cells in tumor tissue have been correlated with nivolumab response; TMB and IFN-γ (GEP) gene expression isrelated to pembrolizumab response. The PRECISION-01 (NCT03917537), a whole-genome study (WGS), aims to identify a biomarker that can predict the response to immunotherapy by analyzing the data of platinum-refractory HNC patients who show a complete pathological response after 4 cycles of nivolumab immunotherapy treatments [13,14,15,16,17,18,19,20,21,22,23,24].

Colton et al. synthesize preclinical and clinical data and highlight the ability of radiotherapy to “modulate” or “reprogram” the tumor microenvironment. Jarosz-Biej and collaborators call the tumor microenvironment as “game-changer” for radiotherapy. A tumor microenvironment rich in myeloid-derived cell populations is the ideal candidate to be manipulated by irradiation, but in this case, radiotherapy has more likely an immunosuppressive role. At the same time, irradiation of several tumor sites is supposed to have a more intense immune antitumor effect, taking into account the diverse repetoire of T-cells. A decisive factor in radio-resistance and progression is tumor stroma, tumor-associated fibroblasts being factors associated with radio-resistance through endothelial cells and adipocytes that modulate angiogenesis. Recently, Ansems et al. introduced the concept of “crosstalk” between cancer-associated fibroblasts, immune cells and tumor cells in establishing radiosensitivity [25,26,27,28].

## 4. Radiotherapy and Immunotherapy for HNC–Clinical Evidences

The first clinical trial that assesed the response of HNC to PD-1 inhibitors was KEYNOTE-012 which showed an overall response rate (ORR) of 18% regardless of HPV status. In the study, there were reported cases in which the response to treatment was maintained > 30 months, the toxicity profile being also favorable. Median overall survival (OS) reported at 12 months was 38%, and the response was maintained > 6 months in 85% of the cases. The KEYNOTE−055 trial evaluated pembrolizumab immunotherapy in HNC platinum and cetuximab pretreated patients that progressed after 6 months. Out of the 171 patients included in the study, 82% were PD-L1 positive, with a combined positivity score (CPS) ≥ 1 and in 22%, HPV status was positive. The rate of adverse effects of grade> 3 was 15%, and ORR was 16%, with a median duration of response of 8 months. No significant outcomes difference between HPV and PD-L1 positive groups wereseen. The study demonstrated thebenefit of immunotherapy with pembrolizumab, even in cases of intensely pretreated HNC patients [4,14,15,16,17,18,19,20,21,22,23,24].

CHECKMATE 141, a phase III trial, included 361 patients randomized in two arms, nivolumab or an agent at the investigator’s choice between Docetaxel, Methotrexate and cetuximab. The trial included patients being considered refractory to platinum treatment. In the case of chemotherapy/anti-EGFR therapy, the median OS was 6.9 months and 8.4 months for immunotherapy. The PFS rate at 6 months is 19.7% with nivolumabversus only 9.9% with standard therapy. However, 80% of the patients receiving immunotherapy do not respond to treatment, but a small percent of the patients will bring a significant benefit and long-term survival.These results justify the identification of biomarkers and the conception of therapeutic combinations in order to convert those patients from immunotherapy “nonresponders” to “responder”. Tumor proportion score (TPS) is a PD-L1 measurement that is also applied in HNC, evaluating only the expression for tumor cells. The CPS considers PD-L1 expression both for tumor cells and immune cells. CPS appears to be a better predictor of immunotherapy response in HNC. The KEYNOTE-048 phase-3 trial randomized 882 participants into groups that received pembrolizumab alone, pembrolizumab plus platinum/5-fluorouracil or the EXTREME regimen (cetuximab with chemotherapy). pembrolizumab plus platinum and 5-fluorouracil was considered an appropriate first-line treatment for recurrent or metastatic HNC and pembrolizumab monotherapy was considered a first-line monotherapy option only in PD-L1 positive cases [5,16,17,18,19,20,21,22,23,24,29,30,31,32,33].

## 5. The “Local”Immune-Enhanced and “Ab-Scopal”Response—Two Sides of the Same Coin

The concept of using radiotherapy, a loco-regional treatment by definition, to generate systemic tumor responses at a distance from the irradiated anatomical region, a phenomenon called “ab-scopal effect”, is older than 50 years. Formenti and Demaria have made significant contributions to demonstrate the mechanism of cell death mediated by the immune system stimulated by irradiation. Considered a “loss of function” cancer immunologically, especially due to the presence of inactivating mutations TP53 and CDKN2A, HNC could benefit from the “activating” immune effect of irradiation. Augmentation of the ab-scopal effect is observed in the case of combination with immunotherapy, especially if a double combination of anti-PD-1/PD-L1 and anti-CTLA-4 is used. Currently, ab-scopal is evaluated in clinical trials, the synergistic effect being identical with that of an in situ vaccine. Golden and collaborators demonstrate the ab-scopal effect by analyzing the data from 41 patients with solid tumors and at least one distant site of metastatic disease. Using irradiation of a lesion with a total irradiation doseof 35Gy in 10 daily fractions over 2 weeks and granulocyte-macrophage colony stimulation factor (GM-CSF), an ab-scopal response rate was observed in 27% of patients, OS being 21 months versus 8 months in favor of the patients who showed ab-scopal response [33,34,35,36,37,38,39].

## 6. Radiotherapy for HNC in the Immunotherapy Era—IsitTime for a New Concept?

Highlighting the phenomenon of imaging pseudo-progression, having as substrate the infiltration of the tumor microenvironment with inflammatory cells, edema and necrosis generated by immunotherapy, brings up to date the need to reevaluate the concept of planning target volumes for the definitive radiotherapy of locally advanced HNC. If immunotherapy has a well-established place in the future in the definitive treatment of locally advanced HNC, the classic clinical target volume (CTV) could be replaced by “immunological-clinical target volume (ICTV)” so as to ensure the eradication of microscopic disease, taking into account the fragile balance between the immunosuppressive potential and the augmentation of the immune effect by the irradiation.

During evolution, approximately 75% of patients with HNC will benefit from curative or palliative radiotherapy. Grewal et al. mention rates of 88% of patients that initially benefited from an irradiation session will need palliative treatment.

In the case of non-surgical treatment of locally advanced HNC, it is unanimously accepted that a dose of 70Gy/35 daily fractions over 7 weeks using the IMRT technique and concurrent chemotherapy with cisplatin is the gold standard. Recently, the results of trials proposing the de-escalation of treatment for HNC HPV+ cancers were published, cisplatin substitution with cetuximab demonstrating inferiority in outcome for this subtype of the disease, without a benefit in reducing the toxicity profile, cetuximab remaining an option reserved for platinum non-eligible patients. In the case of recurrent or metastatic disease in which the goal is palliation of symptoms, multiple therapeutic regimens were proposed. Among these, we mention 20Gy/5 daily fractions/one week, the “spilt course” regimen proposed by Stevens et al. consisting of2 equally cycles of 50Gy2.5Gyper fraction and the weekly regime of 30–36Gy/5–6 fractions. The “quad shot” protocol using the VMAT technique for recurrent/metastatic HNC demonstrated in the 8502 trial, conducted by the Radiotherapy Oncology Group (RTOG), remarkable results with low toxicity rates. The protocol includes 2 daily fractions of 3.7Gy with an interval of at least 6 h for 2 consecutive days, the total dose being 14.8Gy/4 fractions. The protocol was repeated every 3–4 weeks for up to 3 cycles. Tumor response was achieved in 85% of patients with symptom relief in 77% of the cases, the delivery of at least 2 treatment sessions being associated with a favorable response, and a median OS of 5.7 months. The absence of acute or late grade ≥3 toxicities recommends this protocol, confirming the data obtained by Corry and collaborators in a phase II trial [1,2,3,4,5,13,31,39,40,41,42,43,44,45,46,47,48,49,50,51].

Given the clinical and preclinical evidence associating the synergistic effect with an increased fractional dose, but also favorable results in local and symptom control, the “quad shot” protocol and the weekly 6Gy fractional dose protocol could be the preferred option. Although the concept of systemic treatment in recurrent/metastatic setting of HNC is constantly changing with the publication of the results of the KEYNOTE 048 trial, opening new horizons for immunotherapy with pembrolizumab, the only case of the ab-scopal effect associated with “quad shot” radiotherapy for metastatic HNC was reported in a case treated with Ipilimumab, an anti-CTLA-4 monoclonal antibody. The combination of palliative radiotherapy with other immunotherapeutic agents such as durvalumab and tremelimumab, an association that has been shown to be effective for patients treated with stereotactic body radiation therapy (SBRT), should also be investigated in clinical trials to evaluate the “local” or “ab-scopal” synergistic potential [37,42,47,51,52,53,54,55,56,57].

## 7. Re-Irradiation with Immunotherapy-Promising Horizons?

The pattern of therapeutic failure in HNC patients is the high rate of loco-regional recurrence that is the cause of death in most patients. About 40% of long-term survivors will develop the second primary tumor in the head and neck region. Salvage surgery can provide survival rates of up to 40% at 5 years, but there are also situations when surgery is contraindicated or may only be incomplete due to the associated risks. In these situations, re-irradiation as a single method or in combination with chemotherapy, EGFR inhibitor, or PD-1/PD-L1 immune checkpoint inhibitors may be an option. However, given the high rate of potentially severe or fatal toxicities, estimated at approximately 72% at 5 years, including carotid blowout and necrosis, generally associated with a cumulative dose of 128–130Gy, the re-irradiation decision should be taken with precaution. Radiation doses higher than 60Gy are associated with significant benefit and curative potential if IMRT and volumetric modulated arc therapy (VMAT) techniques are used. By generating a steeper dose gradient, these modern techniques reduced the risk of associated toxicities and are associated with an increasing rate of re-irradiation clinical decision. Fibrosis that reduces the potential tumor exposure to chemotherapy and the technical impossibility of re-irradiating bulky tumors without the risk of high-grade toxicity for organs at risk (OARs) are just some of the impediments to implementing re-irradiation in the clinical routine. There is a consensus against elective ganglion irradiation and against the use of expansions >0.5mm from gross tumor volume (GTV) to planning target volume (PTV). In case of re-irradiation after salvage surgery, it is recommended to use setup margins of 0.3–0–5mm from clinical target volume (CTV) to PTV. The CTV includes, in this case, the whole surgical bed. The rate of distant metastases in HNC cancers is relatively low (between 4.0% and 26.0%) in patients who were initially multimodal treated. An option in this case for the synergistic use of immunotherapy and radiotherapy should be the irradiation of metastasis outside the initial irradiation field to obtain the ab-scopal effect. The majority of patients initially treated with chemo-radiotherapy will develop loco-regional recurrences over time. In this situation, re-irradiation with curative or palliative potential must be taken into consideration. A partial tumor volume re-irradiation with high doses per fraction or omission of GTV-CTV expansion should be considered if the risk of severe toxicity, especially the risk of a carotid blowout, is increased. Thus, cases that would not be initially eligible for re-irradiation may benefit from this treatment as an enhancer of the immunotherapy effect, even if the proposed concept of partial-volume irradiation cannot offer tumor control as a single treatment option [1,44,45].

## 8. Conclusions

High PD-L1 expression HPV-radioresistant tumors may be the best candidates for the synergistic effect of RT–IT. For metastatic HNC cases, ab-scopal irradiation and loco-regional recurrences curative re-irradiation by IMRT and VMAT techniques or palliative “quad Shot” in combination with immune checkpoint inhibitors are strategies that can be considered. The synergistic effect of RT–IT has been demonstrated in clinical practice, and for HNC RT–IT, the overall response rate is about 18%. Given the demonstrated potential of radiotherapy to activate the immune system through already known mechanisms, the identification of biomarkers that direct “nonresponders” towards a synergistic stimulation strategy is necessary. With the development of modern techniques, re-irradiation is becoming an increasingly common option for patients who have previously been treated with higher doses of radiation for curative intent. Recurrent and metastatic HNC radiotherapy could evolve from the first level of knowledge, that of ballistic precision, dose conformity and homogeneity, to level two, of “biological dose painting”, in which the concept of tumor heterogeneity and radio-resistance supports the need for escalation of doses on biological criteria. Level three of evolution is thus of “immunological dose painting”, in which the concept of target volumes and a tumoricidal dose can be completely reevaluated. In this case, the target can become even the tumor microenvironment or a partial volume of primary tumor or metastasis, not all the gross and microscopical disease and tumoricidal BED may be completely different from the currently estimated values, radiotherapy treating the tumor in this case indirectly by boosting the immune response.

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
