# Peer review of "Synergies Radiotherapy-Immunotherapy in Head and Neck Cancers. A New Concept for Radiotherapy Target Volumes—“Immunological Dose Painting”"

_medicina, 2020, doi:10.3390/medicina57010006_

Round 1

Reviewer 1 Report

It is appreciated that the authors revised the manuscript and responded to the suggestions. I still like the aspect that irradiation, in some cases, can help to sensitize the tumor for eradication by the immune system. However, to be frank, the manuscript for me appears difficult to read. Firstly, the title still sounds laborious. It would be very helpful if you could shorten and make the title catchier for the general reader. Also, since this is an Opinion article, there is no original research involved. Therefore it is irritating to read, e.g. in lanes 54, 55, 91 "...is investigated…” This sounds like an investigation was done in this report. Please also recheck the manuscript regarding its wordiness. Most of all, I am convinced that this manuscript could benefit from a figure in which the authors clearly and graphically highlight the message of their opinion article (including the 3 levels).

Author Response

Dear reviewer,

Thank you very much for your time and for appreciating the changes. We replaced the title with a more concise one "Synergies radiotherapy-immunotherapy in head and neck cancers - focus on radiotherapy planning" and we replaced that expression "is investigated" with “was evaluated”. We introduced a representation of the three concepts of target volumes. Also, at the suggestion of another reviewer, we introduced a short paragraph and 4 additional bibliographic notes (24-28) on the irradiation of the tumor microenvironment. I hope you will appreciate the new version.

Best regards,

Camil Mirestean

Reviewer 2 Report

Dear authors,

the MS was significantly changes compared to the first version. Within this manuscript a very important topic is discussed. You should refer to a couple of new studies, what happened by irradiation to the tumor microenvironment.

Author Response

Dear reviewer,
Thank you very much for your time and for appreciating the changes. We have introduced a short paragraph and 4 additional bibliographic notes (24-28) on the irradiation of the tumor microenvironment. Also, at the suggestion of another reviewer, I replaced the title with a more concise one "Synergies radiotherapy-immunotherapy in head and neck cancers - focus on radiotherapy planning" and I replaced the expression "is investigated" with "was evaluated". We also introduced a representation of the three concepts of target volumes. I hope you will appreciate the new version.

Best regards,
Mirestean Camil

Reviewer 3 Report

Dear authors, to this reviewer the changes provided are appropriate, and respond to the requests. 

Author Response

Dear reviewer,
Thank you very much for your time and for appreciating the changes. At the suggestion of the other reviewers, we introduced a short paragraph and 4 additional bibliographic notes (24-28) on the irradiation of the tumor microenvironment. We also replaced the title with a more concise "Synergies radiotherapy-immunotherapy in head and neck cancers - focus on radiotherapy planning" and replaced the phrase "is investigated" with "was evaluated". We also introduced a representation of the three concepts of target volumes. I hope you will appreciate the new version.

Best regards,
Mirestean Camil

Round 2

Reviewer 1 Report

The authors reponded to the comments and improved the manuscript further. They also included three figures (pictures?). These 3 figures could also be easily presented as one and should include more information in the legend. Although significantly revised, the text still could be improved since there are a number of difficulties in expression, grammar and syntax which make the manuscript uneasy to read. After thorough proofreading by a native english speaker, the manuscript could eventually become suitable for publication in Medicina.

Author Response

Dear reviewer,

Thanks for the comments on the improvements that can be made to the manuscript. I asked for the help of a colleague who has an international certificate in English to correct mistakes and I revised the picture and the attached explanations. I hope that in this version the image is more explicit.
Best regards,

Camil Mirestean

This manuscript is a resubmission of an earlier submission. The following is a list of the peer review reports and author responses from that submission.

Round 1

Reviewer 1 Report

The Opinion article „Synergies radiotherapy-immunotherapy in head and neck cancers. A new concept "immunological dose painting" submitted to Medicina is of interest to the field of head and neck oncology.

The authors address an interesting aspect in tumor therapy: Can irradiation help to sensitize the tumor for eradication by the immune system?

To underline their opinion, the authors present examples showing that administration of anti-CTLA4 and anti PD-L1 immune checkpoint inhibitors together with radiotherapy could have therapeutic advantages. Here they shortly discuss potential underlying mechanisms such as irradiation-induced generation of novel antigens that can be recognized by the immune system. They also cite trials such as the KEYNOTE-001 trial involving Pembrolizumab administration in conjunction with radiotherapy. However, the respective reference could not be found. Do the authors refer to the article by Shaveridan et al.? (Shaverdian N, Lisberg AE, Bornazyan K, Veruttipong D, Goldman JW, Formenti SC, Garon EB, Lee P. Previous radiotherapy and the clinical activity and toxicity of pembrolizumab in the treatment of non-small-cell lung cancer: a secondary analysis of the KEYNOTE-001 phase 1 trial. Lancet Oncol. 2017 Jul;18(7):895-903. doi: 10.1016/S1470-2045(17)30380-7). Similarly, there are also other references missing in the text, please check!

In summary, the authors point out the potential usefulness of re-irradiation. In particular, they propagate a combination of radio- and immunotherapy to be potentially beneficial for tumors with high PD-L1 expression as well as HPV positive tumors that are resistant to single radiotherapy.

In general, the discussed aspects are interesting. However, the authors need to substantially revise the manuscript before a decision can be made.

Other points:

- The title of the manuscript should be rephrased since it does not sound fully clear.

- It is recommended that the manuscript is proofread by a native English speaker since there are several grammatical errors and typos.

Reviewer 2 Report

  1. Radiotherapy-immunotherapy synergy- from mechanisms to clinical practice

This chapter starts without any introduction. Please insert an introduction.

There are no citations given. Please insert all citated sources.

  1. Radiotherapy and immunotherapy for HNC– clinical evidences

Line 178: 82% were PD-L1 positive: What does PD-L1 positive means? How the authors defined PD-La positive?

The previous studies on the use of PD-L1 as palliative therapy are summarized in this section. But still the question, if the immunotherapy is suitible as radiosensitizer or as addition to radiotherpy is not discussed.

  1. Re-irradiation in the immunotherapy era- it`s time for a new concept?

Line 232: “0.3-0-5mm“

The question of the section heading „Re-irradiation in the immunotherapy era- it`s time for a new concept?“ will not answered within this section. Please discuss the influence of immunotherapy on the radiation therapy. There a multiple studies in progress. Is radiation dose de-escalation realistic,? All de-escalation studies in relation to HPV-status was negative.

Please fill Author Contributions, Funding, Acknowledgments and Conflicts of Interests.

Reviewer 3 Report

Dear authors, the topic presented is highly relevant and innovative. 

This reviewer appreciated the efforts to address novel approaches to the treatment of head and neck tumors. However, the data presented are quite poor, to define the era of an immune dose-painting in HNSCC oncology.

This reviewer would request to widen the discussion on the different RT options, in term of schedules and doses, and better explain how they relate to the immune- response. For example, the authors suggest interestingly 2 different effects of RT on the "local" immune-enhanced and "ab-scopal" response.

Then, it would be useful to explain better how different RT schedules should be associated with different doses of immune-agents. This is outlined but not further developed in the introduction. In addition, the rational behind should be also explained. 

Consider some works for references:

  • DOI:https://doi.org/10.1016/S1470-2045(19)30306-7
  • https://doi.org/10.18632/oncotarget.20760
  • doi: 10.1097/JCMA.0000000000000234
  • and take in consideration: https://ascopubs.org/doi/full/10.1200/EDBK_238339